# Evaluating Food Safety Compliance and Hygiene Practices of Food Handlers Working in Community and Healthcare Settings in Kuwait

**DOI:** 10.3390/ijerph18041586

**Published:** 2021-02-08

**Authors:** Ola H. Moghnia, Vincent O. Rotimi, Noura A. Al-Sweih

**Affiliations:** Department of Microbiology, Faculty of Medicine, Kuwait University, Al-Safat 24923, Kuwait; ola.maghnia@ku.edu.kw (O.H.M.); bunmivr@yahoo.com (V.O.R.)

**Keywords:** food handlers, food safety compliance, hygiene practices, foodborne diseases, Kuwait

## Abstract

Safe food handling and proper hygiene practices performed by food handlers (FHs) in catering establishments are fundamental elements in reducing foodborne diseases. This study aimed at assessing food safety knowledge and compliance of hygiene practices of FHs within food establishments (using a structured questionnaire). A cross-sectional study was carried out from May 2016 to March 2018 on FHs working in community and healthcare settings. A total of 405 FHs, including 44.9% and 55.1%, were working in community and healthcare settings, respectively. The majority, 84.7%, were males with a ratio of 5.5:1. Most of them, 84.4%, had a high school education and above. A greater number, 44%, of FHs were in the age bracket of 29–39 years. As high as 95.6% of them underwent a regular medical check-up. Unsafe attitudes were shown by 44.9%% who used the same hand gloves while handling raw meat and fresh food. Additionally, 42% went home with their uniforms. The hygiene assessment score was 95.8%. In general, FHs have adequate knowledge and compliance with food safety practices. It is recommended that regular and ongoing training on hygienic practices and proper food safety techniques must be given to all FHs to ensure food safety.

## 1. Introduction

Food serves as a source of various pathogens and an excellent vehicle by which many pathogens can reach an appropriate colonization site in a new host. The ingestion of contaminated food containing antibiotic-resistant bacteria may lead to the creation of a reservoir of resistance genes, thereby endangering the health of consumers [1]. Food-borne infections are an escalating global public health concern with significant morbidity and mortality, even in societies with highly industrialized food safety systems [2]. The World Health Organization (WHO) reported that, globally, food-borne diseases affect an estimated 30% of the population in developed countries annually. Occupationally exposed workers such as food handlers (FHs) play an essential role in the transmission of microbial pathogens through food in catering services. According to the WHO in 2007, up to 70% of diarrheal diseases are associated with the consumption of contaminated food, and around 1.8 million people die every year from diarrheal diseases, mostly due to the contamination of food and drinking water [3]. About 76 million cases resulting in 325,000 hospitalizations and 5000 deaths are estimated to occur annually in the USA [4]. Occupationally exposed workers such as FHs are considered an inevitable source of food-borne diseases in the community and healthcare catering services. According to the Codex Alimentarius, FHs are the personnel engaged in the food business [5]; they handle food or items that may come into direct contact with food and utensils not meant for their personal use, involved in making, cooking, serving, transporting, delivering and packing food in any food premises. The chances of food contamination largely depend on their health status and hygiene practices [6]. Therefore, FHs play a vital role in preventing food contamination that may occur at any point in its journey from the producer to the consumer. Previous studies have found that poor personal hygiene could be a potential source of infections and may serve as a reservoir of genes for antimicrobial resistance in organisms [7]. Good hygienic practices of food handlers in the kitchen remain the most effective measure for the prevention of foodborne outbreaks, and food handlers’ attitudes have a crucial impact on their practices [8,9,10]. Studying food handlers’ hygienic practices and association with demographic variables is essential for planning, implementing, and ensuring food safety. There is a paucity of data about the knowledge of food handling practices in food businesses in Kuwait, and it is pertinent that food handling problems need to be addressed using a questionnaire survey which is practical, economical and able to cover a broader range of participants. To this extent, this study was designed to investigate differences in the socio-demographic backgrounds of FHs working in the community (CFHs) and healthcare settings (HCFHs), as well as to assess their knowledge, attitude and practices towards food safety and hygiene. 

## 2. Materials and Methods

### 2.1. Study Design and Population

The target population of this study included food handlers working in community and healthcare settings in the six Governorates of Kuwait. A questionnaire-based face-to-face interview method was performed. The questionnaire took between 15 and 20 min to complete and started with reassuring the participants that the data being collected were confidential in order to increase the likelihood of respondents answering the questions honestly and accurately. 

### 2.2. Study Settings

Kuwait is one of the six Gulf Cooperation Council countries (GCCC) located in the Arabian Peninsula. Kuwait consists of six Governorates; Al-Ahmadi, Al-Asimah, Al-Farwaniya, Hawali, Al-Jahra and Mubarak Al-Kabeer (MAK), each with its own Municipality Food Control/Inspection regional offices. The State Public Health Laboratory routinely screens for the microbial safety of all food items that are imported from different parts of the world. The aim of the Kuwaiti Food Safety and Control System is to safeguard the Kuwaiti food supply with high-quality and safe standards [11]. 

The interviews were carried out among participants’ workplaces and during a routine checkup at the Public Health Department (Food Handler Examination Section), Ministry of Health (MOH). The volunteer FHs received an explanation about the aim of the study and written informed consent was obtained before administering the questionnaire.

### 2.3. Questionnaire Design and Collection 

Quantitative and qualitative evaluations were conducted in assessing the food handlers’ sanitation practices, food-handling knowledge and attitude towards food safety. A written questionnaire was adapted and modified from previously published works [12,13] entailing close-ended questions. Modifications comprise translation from English to the Arabic language in line with local practices. The modified questionnaire was carefully scrutinized for content and expert validation. It was then structured into six sections. Section 1 was about general socio-demographic information of the study population and consisted of five multiple-choice questions: ethnicity, nationality, gender, age, and education level. Section 2 was about work experience in food service, and it included four multiple-choice questions: Governorate of their work area, type of working space, service duration and type of assignments. Section 3 was about risk factors and contained five multiple-choice questions about antibiotic consumption, travel history, compliance with fingernail trimming policy, frequency of ailments and previous hospitalization. Section 4 was about medical practices and comprised of two yes/no questions on food handling certification and compliance with medical check-ups. Section 5 evaluated handwashing practices and included eight yes/no questions, such as the availability of handwashing facilities in the working area and handwashing practices; with water only after visiting the toilet, after using the toilet with both soap and water, before preparing food, after waste handling, after nose-blowing, in-between handling fresh and cooked food, and hand drying thoroughly after handwashing. Section 6 dealt explicitly with respondents’ knowledge of personal hygiene and cross-contamination and sanitation practices, it included five yes/no questions: changing utensils in-between handling fresh food and cooked food; reporting illness; taking work uniform home; wearing the same gloves in-between handling raw meat, fresh vegetables and fruit; and covering cuts and wounds in case of injury. Each of the 15 hygiene practices and food safety knowledge questions in Sections 4, 5 and 6 was coded 0 or 1, with correct responses coded 1 and incorrect responses coded 0. Responses were summed, the score was given for each correct answer and the scores were converted into percentages. A food safety practice score greater than or equal to 67% (10 out of 15 questions) was classified as satisfactory, while those with a score of less than 67% had unsatisfactory status. 

This cross-sectional study was accomplished during the period of May 2016 to March 2017. The study included 405 randomly selected FHs from each district. Data collection was anonymous and began with the informed consent process.

### 2.4. Data Collection and Analysis

The data obtained from the questionnaires were quantitatively tabulated and analyzed using Social Package for Social Scientist (SPSS) software for windows 25.0 (SPSS, Chicago, IL, USA). Results were summarized as mean rank for continuous variables and frequency and percentages for categorical variables. Chi-square test (χ^2^ test) or Fisher exact test were used to test associations between categorical variables. Additionally, the Kruskal–Wallis test was used to test the correlation and attribution between ordinal variables. The Independent Samples *t*-test was used to test for statistically significant differences between the means of two independently sampled groups for interval/ratio variables, while Wilcoxon Mann–Whitney was used in the case of ordinal variables. The two-sided *p*-value was set at 0.05.

## 3. Results

### 3.1. Demographic Characteristics of Study Populations

The comparisons of demographic characteristics, food-handling and hygiene practices between HCFHs operating within hospital premises versus CFHs working in full-service restaurants in the community are highlighted in Table 1, Table 2, Table 3, Table 4, Table 5 and Table 6. As demonstrated in Table 1, of the 405 FHs, 182 (44.9%) were CFHs and 223 (55.1%) were HCFHs. There were 349 (86.2%) non-Arab FHs, out of whom 205 (58.7%) were HCFHs and 144 (41.3%) were CFHs. Southeast Asians represented the highest proportion of FHs. Of the 188 (46.4%) Indian FHs, 140 (74.5%) were HCFHs and 48 (25.5%) were CFHs. Over half, 67 (81.7%), of the 82 Filipinos were CFHs and 15 (18.3%) were HCFHs. A total of 343 (84.7%) were males and 62 (15.3%) were females, with a male: female ratio of 5.5:1. Amongst the HCFHs and CFHs, males were 196 (57.1%) and 147 (42.9%), respectively. The ages of the FHs ranged from 18 to 65 years among the HCFHs and 18–59 among the CFHs. The majority (313: 77.3%) of these were in age brackets 18–28 and 29–39 years. Of these 313, 178 (56.9%) were in the age group 29–39, representing 44% of the total. The difference in the age groups between HCFHs and CFHs was statistically significant (*p* = 0.003). A total of 175 (43.2%) and 167 (41.2%) FHs possessed high school and college degree certificates, respectively. When analyzed according to the place of work, there was a statistically significant difference in education level between HCFHs and CFHs (*p* = 0.001).

### 3.2. Analysis of Workplace Experience 

The workplace experience and tasks of FHs are summarized in Table 2. A total of 294 (72.6%) FHs had >1 year of experience in the food service industry. Of these, 152 (51.7%), 76 (25.9%) and 66 (22.5%) had been working for 1–5, >5–10 and >10 years, respectively, at the same place of work. There were no significant differences in service duration among HCFHs and CFHs (*p* = 0.572). Staff members responsible for handling fresh raw food, cooked food and waiters’ roles were 62 (15.3%), 170 (42%) and 173 (42.7%), respectively. An equal proportion of 94 (55%) and 94 (54.3%), and 76 (45%) and 79 (45.7%) were assigned to handle cooked food and serving among the HCFHs and CFHs, respectively.

### 3.3. Knowledge and Predisposing Factors in the Catering Area

The travel history of the FHs is shown in Table 3. A total of 191 (47.2%) gave a history of travel back home at different periods versus 214 (52.8%) who did not travel during the preceding year. The difference between HCFHs and CFHs who gave a history of travel was statistically significant (*p* = 0.001). A total of 372 (91.9%) trimmed their fingernails once weekly to twice per month. Overall, there was no significant difference in trimming fingernails between HCFHs and CFHs (*p* = 0.678).

Table 4 shows that 345 (85.2%) did not consume any antibiotics during the last year of working as FHs. Only 26 (6.4%) took antibiotics; of these, 12 (46%) and 14 (54%) were HCFHs and CFHs, respectively. The difference between HCFHs and CFHs in antibiotic intake was statistically significant (*p* = 0.039). Only 9 (4%) and 10 (5.5%) of the 223 HCFHs and 182 CFHs, respectively, mentioned having symptoms suggestive of some current disease or infection on the day of data collection. A total of 381 (94.1%) FHs, including 204 (53.5%) HCFHs versus 177 (46.5%) CFHs, gave no history of a previous hospitalization during their careers. There was a significant difference between HCFHs and CFHs (*p* = 0.013). 

### 3.4. Food Safety Knowledge and Hygienic Practice Assessment

The questionnaire dealing with food safety knowledge and hygienic practices was made up of 15 questions with two possible answers: “yes” or “no”. The score of this assessment was calculated out of 15. A total of 388 (95.8%) had satisfactory food safety practice scores. Medicare practices, including possession of food handling training certificates, compliance with medical check-ups, and handwashing compliance assessments, are shown in Table 5. Of the 405 FHs, 354 (87.4%) were certified handlers in food preparation and serving. Three hundred and eighty-seven (95.6%) complied fully with the medical check-up requirement. Of these, 221 (57%) versus 166 (43%) were HCFHs and CFHs, respectively, a difference that attained statistical significance between the two groups (*p* = 0.001).

A total of 401 (99%) reported the availability of handwashing facilities in the kitchens and catering areas, comprising 222 (55%) in the HCFHs’ premises and 179 (45%) in the CFHs’ eating centers. The task of washing hands with only water after visiting the toilet was performed by 240 (59.3%) FHs. Only 165 (40.7%) understood that water only was unsatisfactory for handwashing after visiting the toilet (*p* = 0.001). A total of 403 (99.5%) responded “yes” to washing hands with soap and water after toilet visitations, 386 (95.3%) responded “yes” to whether or not they wash their hands before preparing food and 369 (91.1%) practiced handwashing after handling kitchen waste. FHs who understood the importance of handwashing after blowing their nose were 361 (89.1%). A total of 350 (86.4%) washed their hands when switching between handling raw fresh and cooked foodstuffs. A high proportion, 391 (96.5%), dried their hands thoroughly after handwashing.

Table 6 summarizes those who understood food safety guidelines and hygienic practices. Of the total participants, 265 (65.4%) adhered to the protocol regarding utensils’ changing while handling unwrapped raw or cooked foods and understood that they should be kept apart. A total of 387 (95.6%) were aware of reporting to their managers if they were falling ill before coming to work. A high proportion of FHs, 171 (42.2%), went home with their uniforms. There was a statistically significant difference between the two groups: 71 (41.5%) of HCFHs versus 100 (58.5%) of CFHs (*p* = 0.001). A relatively high number, 182 (44.9%), wore the same gloves when handling raw meat and fresh vegetables and fruit: 83 (45.6%) of HCFHs versus 99 (54.4%) of CFHs (*p* = 0.001). A total of 386 (95.3%) understood the necessity for precautions to be taken when working with skin wounds or covering wounds with bandages or finger cuts.

As demonstrated in Table 7, the education level and work experience had different degrees of impact on the food safety and hygiene practices of food handlers. FHs who had elementary education performed better than those who had received high school and college educations in the following aspects: possession of training certificate (*p* = 0.01), handwashing practices in-between handling fresh and cooked food (*p* = 0.001), and change of utensils while cooking raw and cooked food (*p* = 0.002); nevertheless, they scored significantly higher on taking their work uniform home (*p* = 0.000). High school degree holders used the same gloves for raw meat and fresh food more than others (*p* = 0.05). On the other hand, FHs who had more working experience in the food service industry (>10 years) had lower food safety knowledge in the following practices: washing hands with water only after visiting the toilet (*p* = 0.004) and changing utensils while cooking raw and cooked food.

## 4. Discussion

This study provides an insight into the food handling knowledge and hygienic practices of food handlers. Most FHs were from a non-Arab ethnic group, mostly from South East Asia, particularly India and the Philippines. According to Kuwait’s Public Authority for Civil Information (KPACI) in its official census record in 2019, of the 4.2 million people in Kuwait, two-thirds are expatriates. The food service industry in Kuwait tends to hire expatriates. Our study reflected the fact that more foreign FHs are working in food premises than locals. The preponderance of males over females in our study is similar to studies conducted in India and other countries, which showed that a high number of FHs were males [14,15,16]. This is probably due to the local culture’s influence, where females are not to be seen serving men publicly, apart from the job’s stress-relatedness, which requires a considerable amount of physical ability. The fact that almost all the FHs are non-Kuwaitis highlighted the dislike Kuwaitis have for lowly and menial jobs that attract low salaries.

A significant number of FHs were between 18–28 years old and 29–39 years old, and more workers were significantly older in the HCFH group than the CFH group (*p =* 0.003). This observation is concordant with a study conducted in Kenya, which showed that 75% of recruited FHs were in the 18–30 age group [17]. Worthy of note is that a younger age group of FHs was dominant in Kuwait, a finding that slightly contradicted a study reported by Akabanda et al. in Ghana, who found most of the FHs to be above the age of 30 years [12]. 

The overall level of literacy among the FHs, in our study, was encouragingly high and was much higher than the 90% illiteracy rate reported earlier by Prabhu and Shah in India [15]. Interestingly, paradoxically, the level of college education between CFHs and HCFHs attained a statistical significance of *p* < 0.001. This may be due to the expectation of higher communication skills between employees and consumers in a mixed environment such as ours. It was observed during the interview that a good percentage of HCFHs were monolingual non-English speaking employees, and their command of English was low. Therefore, FHs were assisted by their supervisors in answering the questionnaire. It is presumed that those workers may present food handling training difficulties and communication problems and, accordingly, low impact in performing proper food safety practices [18]. It has been documented that there is a good relationship between literacy and the likelihood of adherence to food safety policies [19]. In our study, it is interesting to note that some aspects were performed better with highly educated FHs, such as not taking work uniform home, than those with a lower degree of education. Nevertheless, those with an elementary degree of education performed better than those who had received high school and college education in the following aspects: possession of a training certificate, handwashing practices in-between handling fresh and cooked food, and changing utensils while cooking raw and cooked food.

In our study, 51.7% of FHs had on-the-job working experience of between 1 and 5 years, a finding that is discordant with a study conducted in Turkey [20], which showed that only 36% of the FHs had work experience of 1 to 5 years. However, it is debatable whether the work experience of FHs alone might be enough for the performance of their duties according to food safety standards unless they have had proper food handling training and experience. This assertion has also been confirmed in a study conducted in Malaysia, which demonstrated that effective and ongoing training on food safety and hygiene must be given to all food service employees to ensure the safety of food provided [21]. It is interesting to note that the high level of work experience by the FHs in this study appears to have negatively influenced adherence to workplace safety issues, such as washing practices with water only after visiting the toilet and the changing of utensils while cooking raw and cooked food.

It is of interest to note that the majority of the FHs were aware of the importance of nail trimming. Therefore, this hygienic practice may not necessarily promote pathogenic organisms’ transmission except if the hand is contaminated by fecal material. Several studies had claimed that the area beneath the fingernails of FHs serves as a vehicle for the transportation of microorganisms from their source to the food and/or directly into the body. Consequently, causing food poisoning and transferring resistant genes, they appear to be more relevant to clinical infective sites [22,23]. In this study, few FHs did not adhere to strict fingernail trimming compared to other studies reporting that a very high number of FHs practiced poor personal hygiene and were carriers of some pathogenic bacteria in their fingernails, posing a threat of bacterial disease transmission [24]. Even with this small number of non-compliant FHs, in our study, the possibility of being at risk of transmitting microbes to the environment and food exists. Other studies have corroborated this speculation and reported that there is a relationship between contaminated fingers of FHs by their feces and consumers’ infection via food processing [7,25,26].

The health status of FHs was generally good: around 95% denied any health problem during the interview and only 5% had comorbidity health complaints associated with nausea and diarrhea. A similar observation was reported in a study carried out by Singh et al., where about 87% of their FHs were healthy [27]. A high percentage of FHs, 95.6%, demonstrated positive attitudes towards reporting illness to their management, unlike a study which reported that 92.6% of their FHs were not aware that exposed skin injury should be covered to avoid food contamination [28]. In general, 95.8% of our FHs had adequate hygienic practice across different stages of the food handling chain. However, this observation differed from a study conducted in Turkey, where their FHs lacked the necessary knowledge and sanitary practices of handling food [20]. Possession of a certificate ensures that FHs have the essential knowledge and skills by food auditors to confirm that all possible food safety risks are covered. There was a highly significant difference in periodical medical check-ups between CFHs and HCFHs (*p* < 0.001). This may be explained by the fact that the HCFHs have better and readily available access to medical facilities. A medical examination should be mandatory before commencing and handling food. In this study, 99% of FHs confirmed the availability of necessary handwashing facilities in the vending site and direct access to municipal tap water. The present study showed a high number of FHs who knew the positive impact of regular and prompt handwashing with both water and soap rather than water alone, reflecting their level of knowledge of the importance of using both water and soap as an effective measure to prevent cross-contamination. 

This study advocates a need for continuous awareness and proper food hygiene practices such as handwashing, sustained training to generate a positive attitude, and encouragement to follow the Hazard Analysis and Critical Control Point (HACCP) system. Future studies should involve active collaboration between the government, public health, veterinary and food safety experts. In addition, a larger population of food handlers should be studied so that the authority can establish a more comprehensive approach to ensuring food safety. Moreover, a study of this nature would help in monitoring trends in the existing diseases and in detecting emerging pathogens, which will in turn ultimately help in developing effective prevention and control strategies. Spreading awareness among consumers, farmers and those raising farm animals would be of great importance in formulating control strategies. It would be interesting to expand this study’s scope to include its correlation with rectal colonization rate and the carriage of multidrug-resistant microorganisms.

## 5. Conclusions

The data generated in this study demonstrated a satisfactory level of knowledge about hygienic practices and succinctly highlighted the compliance with food safety and hygiene practice guidelines among food handlers in the State of Kuwait. However, continuous monitoring and training on safety food handling for FHs are mandatory since there has been a breakdown in some safe food handling procedures. It is apparent from this study that good knowledge of food safety is not always practiced by FHs in the real world. The current findings are useful baseline data to design a comprehensive food safety and quality management system that is essential for planning, implementing and evaluating public food handling practices. 

## Figures and Tables

**Table 1 ijerph-18-01586-t001:** Demographic characteristics of healthcare and community food handlers.

Variable		No. (%) of Food Handlers			
	Item	HCFHs *n* = 223	CFHs *n* = 182	*p*-Value	Total FHs *n* = 405
Ethnicity					
	Arab	18 (32.1)	38 (67.9)	0.0001 *	56 (13.8)
	Non-Arab	205 (58.7)	144 (41.3)		349 (86.2)
Nationality					
	Indian	140 (74.5)	48 (25.5)		188 (46.4)
	Filipino	15 (18.3)	67 (81.7)		82 (20.2)
	Sri Lankan	5 (45.5)	6 (54.5)		11 (2.7)
	Pakistani	0	2 (100)		2 (0.5)
	Nepalese	10 (43.5)	13 (56.5)		23 (5.7)
	Malian	0	1 (100)		1 (0.2)
	Others	53 (54)	45 (56)		98 (24)
Gender					
	Female	27 (43.5)	35 (56.5)	0.06	62 (15.3)
	Male	196 (57.1)	147 (42.9)		343 (84.7)
Age					
	18–28	69 (51)	66 (49)		135 (33.3)
	29–39	86 (48.3)	92 (51.7)		178 (44)
	40–49	52 (71.2)	21 (28.8)		73 (18)
	50–59	15 (83.3)	3 (16.7)		18 (4.4)
	>60	1 (100)	0		1 (0.2)
Mean rank		217.65	185.05	0.003 *	
Educational level					
	Elementary	41 (65)	22 (35)		63 (15.6)
	High school	131 (75)	44 (25)		175 (43.2)
	College	51 (30.5)	116 (69.5)		167 (41.2)
Mean rank		168.23	245.6	0.001 *	

* Statistically significant. HCFHs: Healthcare food handlers; CFHs: Community food handlers.

**Table 2 ijerph-18-01586-t002:** Work experience and type of food-related assignments handled by the healthcare and community food handlers.

Experience/Assignments		Number (%) of Food Handlers			
	Item	HCFHs *n* = 223	CFHs *n* = 182	*p*-Value	Total *n* = 405
Service duration					
	<1 year	68 (62)	42 (38)		110 (27.2)
	1–5 years	75 (49.3)	77 (50.7)		152 (37.5)
	5–10 years	38 (50)	38 (50)		76 (18.8)
	>10 years	41 (62)	25 (38)		66 (16.3)
Mean rank		200.16	206.4	0.572	
Job assignments					
	Raw food	35 (56.5)	27 (43.5)		62 (15.3)
	Cooked food	94 (55)	76 (45)		170 (42)
	Waiter	94 (54.3)	79 (45.7)		173 (42.7)

**Table 3 ijerph-18-01586-t003:** Travel history and frequency of fingernail trimming habits of healthcare and community food handlers.

Variable		Number (%) of Food Handlers			
	Item	HCFHs *n* = 223	CFHs *n* = 182	*p*-Value	Total FHs *n* = 405
Travel history					
	None	138 (64.4)	76 (35.6)		214 (52.8)
	1 month	11 (28.2)	27 (71.8)		38 (9.4)
	3 months	12 (44.4)	15 (55.6)		27 (6.7)
	6 months	32 (62.7)	19 (37.3)		51 (12.6)
	1 year	30 (40)	45 (60)		75 (18.5)
Mean rank		185.63	224.28	0.001 *	
Fingernails trimming					
	Rarely	24 (72.7)	9 (27.3)		33 (8.1)
	Once a month	2 (12.5)	14 (87.5)		16 (4)
	Twice a month	3 (37.5)	5 (62.5)		8 (2)
	Once a week	194 (55.7)	154 (44.3)		348 (85.9)
Mean rank		204.32	201.39	0.678	

* Statistically significant.

**Table 4 ijerph-18-01586-t004:** Risk factors among healthcare and community food handlers.

Variable		Number (%) of Food Handlers			
	Item	HCFHs *n* = 223	CFHs *n* = 182	*p*-Value	Total FHs *n* = 405
Antibiotic intake					
	None	197 (57)	148 (25)		345 (85.2)
	1 week	12 (46)	14 (54)		26 (6.4)
	2 weeks	5 (71.4)	2 (28.6)		7 (1.7)
	3 weeks	3 (42.8)	4 (57.2)		7 (1.7)
	4 weeks	6 (30)	14 (70)		20 (4.9)
Mean rank		196.31	211.20	0.039 *	
Morbidity					
	Healthy	214 (55.4)	172 (44.6)		386 (95.3)
	Diarrhea/Vomiting	4 (57)	3 (43)		7 (1.7)
	Stomach pain/Nausea	5 (41.6)	7 (58.4)		12 (3)
Hospital admission					
	Healthy	204 (53.5)	177 (46.5)		381 (94.1)
	<6 months	10 (71.4)	4 (28.5)		14 (3.5)
	>6 months	0	1 (100)		1 (0.2)
	>1 year	9 (100)	0		9 (2.2)
Mean rank		208.33	196.47	0.013 *	

* Statistically significant.

**Table 5 ijerph-18-01586-t005:** Medicare practices and handwashing compliance assessment among healthcare and community food handlers.

Practices	Response	No. (%) of Food Handlers			
		CFHs *n* = 182	HCFHs *n* = 223	*p*-Value	Total FHs *n* = 405
Possession of certificate					
	**Yes**	**189 (53.3)**	**165 (46.7)**	0.075	**354 (87.4)**
	No	34 (66.6)	17 (33.4)		51 (12.6)
Periodical medical check-up					
	**Yes**	**221 (57)**	**166 (43)**	0.001 *	**387 (95.6)**
	No	2 (11)	16 (89)		18 (4.4)
Facility for handwashing					
	**Yes**	**222 (55)**	**179 (45)**	0.330	**401 (99)**
	No	1 (25)	3 (75)		4 (1)
Handwashing practices with water only after visiting the toilet					
	Yes	112 (46.6)	128 (53.4)	0.001 *	240 (59.3)
	**No**	**111 (67.3)**	**54 (32.7)**		**165 (40.7)**
Handwashing practices after using the toilet with both soap and water					
	**Yes**	**223 (55)**	**180 (45)**	0.201	**403 (99.5)**
	No	0	2 (100)		2 (0.5)
Handwashing practices before preparing food					
	**Yes**	**213 (55)**	**173 (45)**	0.827	**386 (95.3)**
	No	10 (52.6)	9 (47.4)		19 (4.7)
Handwashing practices after waste handling					
	**Yes**	**202 (54.7)**	**167 (45.2)**	0.679	**369 (91.1)**
	No	21 (58.3)	15 (41.7)		36 (8.9)
Handwashing practices after nose blowing					
	**Yes**	**196 (54.2)**	**165 (45.7)**	0.373	**361 (89.1)**
	No	27 (61.4)	17 (38.6)		44 (10.9)
Handwashing practices in-between handling fresh and cooked food					
	**Yes**	**190 (54.3)**	**160 (45.7)**	0.428	**350 (86.4)**
	No	33 (60)	22 (40)		55 (13.6)
Hand drying thoroughly after handwashing					
	**Yes**	**217 (55.5)**	**174 (44.5)**	0.35	**391 (96.5)**
	No	6 (42.9)	8 (57.1)		14 (3.5)

* Statistically significant. The correct responses are highlighted in bold.

**Table 6 ijerph-18-01586-t006:** Food safety and hygienic practices among healthcare and community food handlers.

Practices	Response	No. (%) of Food Handlers			
		CFHs *n* = 182	HCFHs *n* = 223	*p*-Value	Total FHs *n* = 405
Change of utensils while cooking raw and cooked food					
	**Yes**	**143 (54)**	**122 (46)**	0.541	**265 (65.4)**
	No	80 (57)	60 (43)		140 (34.6)
Reporting illness					
	**Yes**	**214 (55)**	**173 (45)**	0.659	**387 (95.6)**
	No	9 (50)	9 (50)		18 (4.4)
Taking work uniform home					
	Yes	71 (41.5)	100 (58.5)	0.001 *	171 (42.2)
	**No**	**152 (65)**	**82 (35)**		**234 (57.8)**
Using same gloves for raw meat and fresh food					
	Yes	83 (45.6)	99 (54.4)	0.001 *	182 (44.9)
	**No**	**140 (62.7)**	**83 (37.3)**		**223 (55.1)**
Covering wound in case of injury					
	**Yes**	**213 (55)**	**173 (45)**	0.827	**386 (95.3)**
	No	10 (52.6)	9 (47.4)		19 (4.7)

* Statistically significant. The correct responses are highlighted in bold.

**Table 7 ijerph-18-01586-t007:** Attribution of food safety and hygienic practices to educational level and work experience among healthcare and community food handlers.

Construct	Educational Level				Work Experience				
	Elementary (*n* = 63)	High School (*n* = 175)	College (*n* = 167)	*p*-Value	<1 Year (*n* = 110)	1–5 Years (*n* = 152)	5–10 Years (*n* = 76)	>10 Years (*n* = 66)	*p*-Value
	Mean Rank				Mean Rank				
Possession of certificate	225.71	201.80	195.69	0.010 *	213.73	195.61	206.24	195.36	0.130
Periodical medical check-up	203.64	199.79	206.13	0.371	204.52	202.80	201.47	199.62	0.891
Facility for handwashing	201.00	203.31	203.43	0.689	202.34	201.83	203.16	203.56	0.931
Handwashing practices with water only after visiting the toilet	216.93	204.97	195.68	0.333	219.66	186.95	186.95	227.62	0.004 *
Handwashing practices after using the toilet with both soap and water	205.21	202.00	203.21	0.296	203.34	202.83	201.50	201.50	0.761
Handwashing practices before preparing food	209.57	203.91	199.56	0.268	204.02	199.64	206.29	202.18	0.702
Handwashing practices after waste handling	197.86	202.36	205.61	0.649	210.21	196.46	200.45	205.92	0.265
Handwashing practices after nose blowing	200.29	208.77	197.98	0.267	204.37	203.09	204.42	195.80	0.819
Handwashing practices in-between handling fresh and cooked food	226.93	206.74	190.05	0.001 *	198.87	198.92	196.26	223.97	0.054
Hand drying thoroughly after handwashing	205.64	200.63	204.49	0.52	201.01	202.14	203.47	204.68	0.924
Change of utensils while cooking raw and cooked food	232.64	209.37	185.14	0.002 *	195.44	178.18	212.74	258.48	0.000 *
Reporting illness	203.64	200.94	204.91	0.674	202.68	200.14	212.11	196.56	0.12
Taking work uniform home	224.21	221.39	175.73	0.000 *	201.69	198.96	189.66	226.79	0.15
Using same gloves for raw meat and fresh food	187.93	216.47	194.57	0.05 *	198.01	205.79	197.82	207.80	0.87
Covering wound in case of injury	203.14	203.91	201.99	0.917	205.85	204.96	195.66	199.12	0.32

* Statistically significant.

## Data Availability

Not applicable.

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
