# Peer review of "Evaluating Food Safety Compliance and Hygiene Practices of Food Handlers Working in Community and Healthcare Settings in Kuwait"

_ijerph, 2021, doi:10.3390/ijerph18041586_

Round 1
Reviewer 1 Report
I think that the broad subject of the paper is very interesting. The issue is very important for policies in developing and developed countries as well.
But the paper lacks key pieces of information and presents some inaccuracies.
The research question is not clear. Is it just the comparison between CFH and HCFH? I would suggest to exploit such interesting data with an analysis of the social determinants of the behaviour of FH. For instance, an analysis of the different conducts of FH according to their level of education and service duration would be much more interesting.
More in detail:
- The last paragraph stating the research question is not very clear. Is the aim of the paper a mere description of the sample?
- Materials and methods. Section 2.1 and 2.2 use the same exact words as in “Food safety knowledge, attitude and self‐reported practices among food handlers in Sohag Governorate, Egypt” (Ahmed Fathy Hamed and Nesreen A. Mohammed). Please rewrite them.
Moreover these sections do not provide all the information needed. They should be reunited in one sub-section, which should report at least the total numerosity of the sample.
- Results. Table 1. I suggest to show only the six nationality more numerous.
Table 2. It would be much clearer to present less detailed data.
In the first column: "Collage" I think you meant “College” or University education.
More in general, I suggest to abandon from now on the presentation of data by Governorate and maintain the results on the comparison between HCFH and CFH using table like the following:
|
|
HCFH |
CFH |
p-value |
Total |
|
Gender - - |
|
|
|
|
|
Age - - |
|
|
|
|
|
Educational status - - - College |
|
|
|
|
Section 3.2: like in the previous section.
Section 3.3: in tables 4,5,6,7,8 I suggest dropping the presentation by Governorate and to avoid presenting row percentages. The tables and comments should present columns' shares like in Akabanda et al. 2017. This in order to provide useful information on demographic and social information on Food Handlers’ behaviours and to allow policies aimed to ensure food safety.

Author Response
Dear,
Kindly see the attachment
Regards

Reviewer 2 Report
In this study authors presented interesting findings regarding safety and Hygiene attitudes of food handlers in Kuwait. The fact that 55.1% exhibit unsafe attitudes during food preparation proves the need for further research.
Introduction is adequate providing all the necessary information.
Material and methods are well-presented. I would like to know more about the statistical analysis and the test-selection.
The results section contains 8 rather large tables, presenting in detail all the findings regarding the participants and their attitudes. However, as it is imperative to present all data, I would not suggest on reducing them.
Discussion focuses on the presentation of their findings in comparison with other studies. I would suggest for you to comment more on the significance of your study for future research. For instance, you could add a small paragraph for future perspectives or improvements and suggestions for similar research to other countries. I found quite useful and astute your conclusion sentence to expand this study's scope to include its correlation with the rectal carriage of multi drug-resistant microorganisms.
Author Response
February 2, 2021
Subject: manuscript ijerph-1093501
Dear Dr. Luan,
Thank you for your letter and for giving us the opportunity to submit a revised draft of the manuscript. We appreciate the time and effort that the reviewers have dedicated to reviewing our paper and providing feedback. Their insightful comments have been immensely helpful. We have read all comments by the reviewers carefully and addressed them point-by-point in the manuscript and covering letter. Our responses are highlighted in bold below and all suggested modifications have been incorporated and highlighted in red within the manuscript.
Reviewer 2 comments to the authors:
We also thank this reviewer for his/her positive suggestions
- “In this study authors presented interesting findings regarding safety and Hygiene attitudes of food handlers in Kuwait. The fact that 55.1% exhibit unsafe attitudes during food preparation proves the need for further research. Introduction is adequate providing all the necessary information”.
Author response: Thank you very much for agreeing with us to the objectives of this study.
- “Material and methods are well-presented. I would like to know more about the statistical analysis and the test-selection”.
Author response: Thank you for this encouraging comments. We have added the suggested content to the revised manuscript on page 3, lines 137-149 under section 2.4. Data collection and analysis.
- “The results section contains 8 rather large tables, presenting in detail all the findings regarding the participants and their attitudes. However, as it is imperative to present all data, I would not suggest on reducing them”.
Author response: Thanks for your comment. We agree with you that there is no need to reduce the number Tables. However, we had to comply with the changes suggested by Reviewer 1 who requested us to input the results [n the Tables according to comparison between HCFH and CFH without Governorates.
- “Discussion focuses on the presentation of their findings in comparison with other studies. I would suggest for you to comment more on the significance of your study for future research. For instance, you could add a small paragraph for future perspectives or improvements and suggestions for similar research to other countries. I found quite useful and astute your conclusion sentence to expand this study's scope to include its correlation with the rectal carriage of multi drug-resistant microorganisms”.
Author response: We think this is an excellent suggestion. We have added a paragraph on page 3 of discussion, lines 461-473. It would be interesting to explore this aspect and we intend to investigate the fecal carriage of CRE and MDR bacteria among FHs in another research project.
We hope the revised manuscript meet your high standards and will better suit the International Journal of Environmental Research and Public Health. The authors welcome further constructive comments if any. We thank you for your interest in our research.
Sincerely,
Noura Al Sweih, MD
Round 2
Reviewer 1 Report
The paper has been significantly improved and can be now be published in IJERPH after minor revisions.
Table 4: Mean rank missing for morbidity.
Author Response
Dear,
Comments and Suggestions for Authors
The paper has been significantly improved and can be now be published in IJERPH after minor revisions.
Table 4: Mean rank missing for morbidity.
Author response: Thank you very much. In Table 4, variable as morbidity, including the following: (Healthy, Diarrhea / Vomiting, Stomach pain /Nausea) that are binary variables, and there is no order between them. Therefore, mean rank is not applicable in this case.
Sincerely,
Noura Al Sweih, MD